# Second week methyl-prednisolone pulses improve prognosis in patients with severe coronavirus disease 2019 pneumonia: An observational comparative study using routine care data

**Guillermo Ruiz-Irastorza**[1,2,3]*, **Jose-Ignacio Pijoan**[2,4,5], **Elena Bereciartua**[2,3,6],
**Susanna Dunder**[2,7], **Jokin Dominguez**[2,7], **Paula Garcia-Escudero**[2,8],
**Alejandro Rodrigo**[2,7], **Carlota Gomez-Carballo**[2,7], **Jimena Varona**[2,7], **Laura Guio**[2,3,6],
**Marta Ibarrola**[2,3], **Amaia Ugarte**[1,2], **Agustin Martinez-Berriotxoa**[2,3,7], **On behalf of the
Cruces COVID Study Group**[¶]

**1** Autoimmune Diseases Research Unit, Service of Internal Medicine, Hospital Universitario Cruces,
Barakaldo, Bizkaia, Spain, **2** Biocruces Bizkaia Health Research Institute, Barakaldo, Bizkaia, Spain,
**3** University of the Basque Country (UPV/EHU), Leioa, BI, Spain, **4** Clinical Epidemiology Unit, Hospital
Universitario Cruces, Barakaldo, Bizkaia, Spain, **5** CIBER of Epidemiology and Public Health (CIBERESP),
Madrid, Spain, **6** Infectious Diseases Unit, Hospital Universitario Cruces, Barakaldo, Bizkaia, Spain,
**7** Service of Internal Medicine, Hospital Universitario Cruces, Barakaldo, Bizkaia, Spain, **8** Service of
Rheumatology, Hospital Universitario Cruces, Barakaldo, Bizkaia, Spain

¶ Membership of the Cruces COVID Study Group is provided in the Acknowledgments.
* r.irastorza@outlook.es

University Hospital, GERMANY

## Abstract

### Objective

To analyze the effects of a short course of methyl-prednisolone pulses (MP) during the second week of disease (week-2) in patients with severe coronavirus disease 2019 (COVID-19) pneumonia.

### Methods

Comparative observational study using data collected from routine care at Hospital Universitario Cruces, Barakaldo, Bizkaia, Spain in patients with COVID-19 pneumonia. We compared patients who received week-2-MP (125–250 mg/d x3) with those who did not, with the end-points time to death and time to death or endotracheal intubation.

### Results

We included 242 patients with COVID-19 pneumonia and elevated inflammatory markers at admission. Sixty-one patients (25%) received week-2-MP. Twenty-two patients (9%) died and 31 (12.8%) suffered death or intubation. The adjusted HRs for death and death or intubation for patients in the week-2-MP group were 0.35 (95%CI 0.11 to 1.06, p = 0.064) and 0.33 (95%CI 0.13 to 0.84, p = 0.020), respectively. These differences were specifically seen in the subcohort of patients with a SpO2/FiO2 at day 7 lower than 353 (adjusted HR 0.31,

**Data Availability Statement:** All relevant data are within the manuscript and its Supporting Information files.

**Funding:** The authors received no specific funding for this work.

**Competing interests:** The authors have declared that no competing interests exist.

95% CI 0.08 to 1.12, p = 0.073 and HR 0.34, 95%CI 0.12 to 0.94, p = 0.038, respectively) but not in patients with higher SpO2/FiO2. Patients receiving out-of-week-2-MP, non-pulse glucocorticoids or no glucocorticoids had an increased adjusted risk for both outcomes compared with week-2-MP group: HR 5.04 (95% CI 0.91–27.86), HR 10.09 (95% CI 2.14–47.50), HR 4.14 (95% CI 0.81–21.23), respectively, for death; HR 7.38 (95% CI 1.86–29.29), HR 13.71 (95% CI 3.76–50.07), HR 3.58 (95% CI 0.89–14.32), respectively, for death or intubation. These differences were significant only in the subgroup with low SpO2/FiO2.

## Conclusions

Week-2-MP are effective in improving the prognosis of patients with COVID-19 pneumonia with features of inflammatory activity and respiratory deterioration entering the second week of disease. The recognition of this high-risk population should prompt early use of MP at this point.

## Introduction

Beginning in December 2019, a novel coronavirus, designated SARS-CoV-2, has caused an international outbreak of respiratory illness termed COVID-19 [1, 2]. Glucocorticoids were banned in the initial recommendations for treating the disease [3]. However, severe lung and systemic inflammation may take place usually during the second week of disease, being the main cause of admission to intensive care units, need for mechanical respiratory support and death [4].

Admission of patients with COVID-19 started at Hospital Universitario Cruces in March 2020, reaching its peak between the 27th of March and early April. An initial period of no use of glucocorticoids [3] was followed by the administration of non-pulse glucocorticoids and later methyl-prednisolone pulses (MP) to selected patients. On April 3rd 2020, a joined protocol by the services of Infectious Diseases and Internal Medicine included MP in certain scenarios (see Methods).

The aim of this study is to analyse the effects of MP on the clinical course of patients with COVID-19 pneumonia admitted to the services of Infectious Diseases and Internal Medicine of Hospital Universitario Cruces. Our working hypothesis was that a short course of MP administered within the second week after the onset of symptoms (week-2-MP) would improve the outcome of patients with COVID-19 pneumonia with markers of inflammation and signs of progressive respiratory deterioration.

## Methods

### Study design and population

We conducted an observational study using data collected from routine care to assess the efficacy of week-2-MP to treat patients with COVID-19 pneumonia. The protocol was approved by the Basque Country Research Ethics Committee (code EPA2020032) in accordance with the Declaration of Helsinki's guidelines for research in humans. This study has been registered in the EU PAS Register with the number EUPAS36287.

All patients admitted between March 1st and April 30th to the services of Infectious Diseases and Internal Medicine, Hospital Universitario Cruces, with a diagnosis of COVID-19

pneumonia, supported by chest X-ray and/or CT scan informed by skilled radiologists. All cases were confirmed by positive reverse-transcriptase–polymerase-chain-reaction (RT-PCR) assay for SARS-CoV-2 in nasopharyngeal swabs, were initially selected for the study. Patients were excluded if they had no inflammatory markers at admission (see Therapeutic protocols), died within the first week of symptoms, were admitted to hospital after the end of the second week, died of causes non-related to COVID-19 infection or were initially admitted to the Intensive Care Unit.

## Therapeutic protocols

Initial management included supportive therapy with fluids and oxygenation with the goal of an O2 saturation ≥92%. The antiviral protocol was developed based on the recommendations by the Spanish Ministry of Health and the Spanish Agency for Medicines and Health Products (AEMPS) in the weeks prior to April 3, 2020 [5, 6].

Hydroxychloroquine for 5 days associated with lopinavir/ritonavir for 7–10 days were recommended for all patients with pneumonia. Selected patients with severe pneumonia -CURB65 ≥2 [7] and/or SpO2 in ambient air O2 <90%- could be treated with remdesivir for 5 to 10 days, at the discretion of the clinician, after enrolment in a clinical trial [8] or as an off-label drug. Following the recommendations of the World Health Organization (WHO) [3], glucocorticoids were not used during the initial weeks of the pandemic, unless needed for comorbid conditions.

After the recognition of the inflammatory phase of COVID-19 pneumonia [4], glucocorticoids were empirically given to selected patients with severe disease and positive inflammatory markers, defined as any of the following: lymphocyte count <800/mm3 (normal 1300–2900), platelet count <150000/mm3 (normal 135000–150000), ferritin >1000 ng/ml (normal 15–300), C-reactive protein >100 mg/l (normal 0–11), D-dimer >1000 ng/ml (normal 0–500). This was an empiric definition, in which the levels of ferritin, CRP and D-dimers well above the upper normal limit were set taking into account the marked elevation in such parameters seen in many patients with COVID-19 pneumonia.

Glucocorticoids were initially given at doses around 1 mg/Kg/d during several days and later as MP, 125 to 250 mg/d for 3 consecutive days, similar to the scheme used in our patients with systemic autoimmune diseases [9]. Our therapeutic protocol was updated on April 3rd 2020, including the recommendation of MP for patients with COVID-19 pneumonia with altered/worsening inflammatory parameters (lymphopenia, thrombocytopenia, rising ferritin, D-dimers and or C-reactive protein) and clinical deterioration, particularly those showing impending respiratory failure with decreasing SpO2/FiO2 values. MP were encouraged to be given during the second week after the onset of symptoms, always according to the attending physician best judgment. This therapeutic protocol remained unchanged until the end of the study period.

## Study variables

All the clinical data were extracted from Hospital Universitario Cruces electronic medical records by the study investigators. Databases were accessed to extract the study data between 11th and 30th May 2020. All data were fully anonymized before the analysis and the Basque Country Research Ethics Committee waived the requirement for informed consent.

Baseline variables included age, gender, previous diagnosis of diabetes mellitus, obesity (body mass index ≥30), arterial hypertension, chronic pulmonary disease, active neoplastic, neurodegenerative disease or systemic autoimmune disease and immunosuppressive therapy. The CURB65 scale [7] was calculated at admission and divided into three categories (low,

medium and high risk). In order to assess severity within the second week of disease, the SpO2/FiO2 at that point was included in the database. The definition of high inflammatory state at admission described in the therapeutic protocol in use (see above) was assumed for the purposes of this study.

Therapeutic variables included the following: lopinavir/ritonavir, hydroxychloroquine, non-pulse glucocorticoids (including the average daily dose and the number of days of treatment), low molecular weight heparin and MP (including the week of administration, counting after the onset of symptoms).

## Outcomes

We used two primary outcomes: time to death (attributed to COVID-19) and time to death or endotracheal intubation.

## Statistical analysis

Mean and standard deviation or median and interquartile range were used to describe continuous variables, according to their distributional characteristics. Counts and relative frequencies describe categorical variables. The time of onset of symptoms (fever and/or persistent cough), as reported by the patient, was the time origin for calculation of time to events. Life tables were used to describe risk of events over the follow-up period, which ended either at the occurrence of the event of interest, at discharge or at the end of the study follow-up on 20th May 2020. Kaplan-Meier failure curves with log-rank test were calculated for each of the two primary outcomes, using the dichotomous variable week-2-MP, yes/no, for group comparisons.

Adjusted Cox proportional risk models were fitted to assess the effect of the dichotomous variable week-2-MP as the predictor of main interest. The aforementioned variables were included in the full model. Likelihood ratio tests were used in a sequential fashion in order to find a reduced adjusted model containing statistically significant covariates. Hazard ratios (HR) with 95% confidence intervals were used to estimate the magnitude of the association between risk predictors and outcomes. Proportionality of hazards was tested through the use of comparison of adjusted and predicted survival curves and Schoenfeld residuals.

All the univariate and multivariate analyses were repeated after stratification by the median SpO2/FiO2 at week 2 (equal to 353), in two groups: SpO2/FiO2 ≤353 and >353 (designated as low SpO2/FiO2 and high SpO2/FiO2), as this variable showed a significant departure from the proportionality of hazards assumption.

Finally, the Cox analysis was repeated using a four-category predictor in order to better illustrate the effects of glucocorticoids according to the dose, duration and time of administration. Patients were divided into 4 groups: a) no-glucocorticoids, i.e. patients not receiving glucocorticoids in any form (n = 122); b) non-pulse glucocorticoids, i.e. patients receiving glucocorticoids at doses lower than 100 mg/d for periods longer than 3 days (n = 36, with 10 of them also receiving pulses); c) out-of-week-2-MP, i.e. MP at week 1 or 3, with no additional glucocorticoids at lower doses (n = 30); d) week-2-MP, i.e. patients receiving MP during week 2, with no additional glucocorticoids at lower doses (n = 54). As our proposed schedule consists of MP with no following tapering scheme, we decided to group all patients receiving non-pulse courses of glucocorticoids for longer than 3 days into the same category and keeping in the MP groups those patients receiving only pulses.

Stata 16.1 for Windows was used for all the analyses (StataCorp. 2019. Stata Statistical Software: Release 16. College Station, TX: StataCorp LLC.).

### Patient and public involvement

Neither patients nor the public were involved in the conception or conduct of the study.

## Results

Three hundred and forty-three patients with COVID-19 pneumonia were initially identified. Of these, 252 had an inflammatory state at admission; after excluding 2 patients with early death within the first week of disease course, 2 patients dying with COVID-19 but with death attributable to terminal cancer and 6 patients admitted after the end of the second week of disease, 242 patients were selected for the analysis of the primary and secondary outcomes.

Sixty-one patients (25%) received week-2-MP. In addition, 33 patients (14%) received out-of-week-2-MP (week 1 or 3). The remaining 148 patients (61%) did not receive MP.

Table 1 shows the clinical characteristics of the whole cohort and according to whether or not patients received week-2-MP.

### Outcome: Death

Twenty-two patients (9.1%) died during the study period. The proportion of deceased patients was lower in the week-2-MP group: 4/61 (6.6%) vs. 18/181 (9.9%). The Kaplan-Meier failure curves (Fig 1A) showed a non-significant decreased risk of death of patients in the week-2-MP group (log-rank test, p = 0.102).

The final Cox model showed an adjusted HR for death of 0.35 (95%CI 0.11 to 1.06, p = 0.064) for patients in the week-2-MP group. Other independent predictors of death included a previous diagnosis of arterial hypertension, the use of non-pulse glucocorticoids, a high-risk CURB65 category and SpO2/FiO2 at week 2 (Table 2).

In the subgroup with low SpO2/FiO2, 3/42 (7.1%) week-2-MP patients died vs. 14/80 (17.5%) non week-2-MP patients. The Kaplan-Meier failure curves depicted on Fig 1B showed a significantly lower mortality among week-2-MP patients (long-rank test, p = 0.041).

The final Cox model in this subcohort showed a protective effect of week-2-MP (HR 0.31, 95% CI 0.08 to 1.12, p = 0.073), with the rest of independent predictors being unchanged from the whole cohort model.

No differences were seen in the univariate analysis between patients with or without week-2-MP in the subpopulation with high SpO2/FiO2. Only 5/120 (4%) patients died in this subgroup. The multivariate Cox model could not identify any significant predictor of the outcome.

### Outcome: Death or intubation

Thirty-one patients (12.8%) suffered death or intubation. Week-2-MP patients had a lower incidence of the combined outcome: 6/61 (9.8%) vs. 25/181 (13.8%). The Kaplan-Meier failure curves (Fig 2A) showed a non-significant largest time free of events in patients receiving week 2 MP, (log-rank test p = 0.125).

In the final Cox model, the adjusted HR for week-2-MP was 0.33 (95%CI 0.13 to 0.84, p = 0.020). The same independent predictors than in the model with death as the outcome variable, with the exception of arterial hypertension, were retained (Table 3).

In the subcohort with low SpO2/FiO2, the combined outcome was met by 5/42 (12%) vs. 20/80 (25%) patients receiving and not receiving, respectively, week-2-MP. The failure curves on Fig 2B showed a significantly better outcome in the former group (log-rank test, p = 0.032).

In the final Cox model, week-2-MP were beneficial (HR 0.34, 95%CI 0.12 to 0.94, p = 0.038). The model retained the same final variables than the whole cohort analysis.

**Table 1. Clinical characteristics of the cohort according to treatment with MP.**

| VARIABLE | Overall (n = 242) | Week-2-MP (n = 61) | No week-2-MP (n = 181) | p-value[1] |
|---|---|---|---|---|
| Age (years), mean (sd) | 64.4 (14.3) | 65.0 (12.1) | 64.2 (15.0) | 0.702 |
| Male n (%) | 150 (62.0) | 40 (65.6) | 110 (60.8) | 0.504 |
| Follow-up* (days), mean (sd) | 17.9 (7.3) | 20.7 (6.0) | 17.0 (7.5) | **0.0007** |
| Diabetes mellitus n (%) | 51 (21.1) | 9 (14.8) | 42 (23.2) | 0.162 |
| Overweight n (%) | 49 (20.3) | 10 (16.4) | 39 (21.6) | 0.386 |
| Hypertension n (%) | 117 (48.4) | 33 (54.1) | 84 (46.4) | 0.299 |
| Chronic bronchopathy n (%) | 62 (25.6) | 14 (23.0) | 48 (26.5) | 0.581 |
| Active cancer n (%) | 26 (10.7) | 7 (11.5) | 19 (10.5) | 0.831 |
| Neurodegenerative disease n (%) | 9 (3.7) | 3 (4.9) | 6 (3.3) | 0.567 |
| Autoimmune disease n (%) | 9 (3.7) | 3 (4.9) | 6 (3.3) | 0.567 |
| Immunosuppressive therapy n (%) | 21 (8.7) | 5 (8.2) | 16 (8.8) | 0.877 |
| CURB65, high risk n. (%) | 19 (7.9) | 5 (8.2) | 14 (7.7) | 0.461 |
| Time of symptoms to admission (days) mean (sd) | 6.6 (3.2) | 7.4 (2.8) | 6.3 (3.2) | **0.021** |
| Lymphocytes (count/mm$^3$) median (iqr) | 800 (580) | 680 (480) | 815 (575) | 0.118 |
| Platelets (count/mm$^3$) median (iqr) | 212,000 (144,000) | 208,000 (156,000) | 213,000 (135,500) | 0.703 |
| Ferritin (mg/dl) median (iqr) | 543 (807) | 824 (919) | 481 (723) | **<0.001** |
| D-dimers (ng/ml), Median (iqr) | 500 (1,171) | 540 (870) | 470 (1,171) | 0.224 |
| C-reactive protein (mg/dl) median (iqr) | 79.5 (112.8) | 112.3 (89.8) | 73.4 (99.9) | **0.024** |
| SpO2/FiO2, median (iqr) | 380 (160.0) | 332 (201.0) | 438 (125.0) | **<0.001** |
| SpO2/FiO2 < 353 n (%) | 117 (48.3) | 41 (67.2) | 76 (42.0) | **0.001** |
| Hydroxychloroquine n (%) | 224 (92.9) | 61 (100) | 163 (90.5) | **0.013** |
| Days on hydroxychloroquine mean (sd) | 6.25 (2.6) | 6.8 (2.6) | 6.1 (2.6) | 0.24 |
| Lopinavir/ritonavir n (%) | 218 (90) | 58 (95) | 160 (88) | 0.13 |
| Betaferon n (%) | 13 (5.3) | 1 (1.6) | 12 (6.6) | 0.13 |
| LMWH n (%) | 225 (92.9) | 60 (98.3) | 165 (91.26) | **0.057** |

Week-2-MP: methyl-prednisolone pulses in week 2; LMWH: low molecular weight heparin; sd: standard deviation; iqr: interquartile range.

* From disease onset to death, discharge or end of the study period.

Six patients suffered the combined outcome in the subcohort of patients with high SpO2/FiO2. Again, no differences were seen between patients with or without week-2-MP in the univariate or multivariate analysis. However, the use of non-pulse glucocorticoids was associated with an increased risk for the combined outcome (HR 10.5, 95%CI 0.94 to 116.84, p = 0.056).

## Analysis of the four-level glucocorticoid variable

Table 4 depicts the clinical characteristics of the four subgroups.

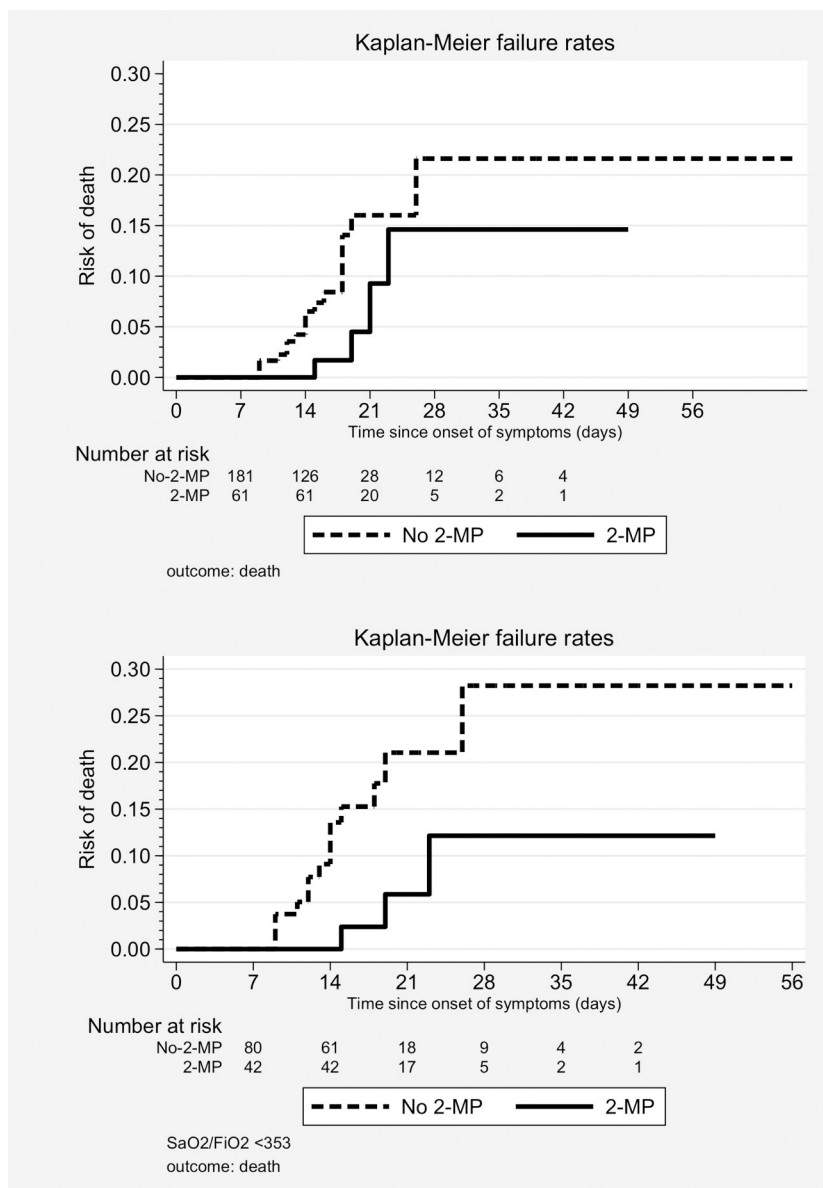

**Fig 1. Kaplan-Meier failure curves, second week methyl-prednisolone pulses (2-MP) vs. no 2-MP.** Outcome: death. (a) Whole cohort (n = 242). Log-rank test, p = 0.102. (b) Patients with low SpO2/FiO2 (n = 122). Log-rank test, p = 0.041.

In the model with mortality as single outcome, week-2-MP patients showed lower mortality (HR 0.24, 95%CI 0.05 to 1.24, p = 0.088), whilst non-pulse glucocorticoid patients had an increased risk of death (HR 2.44, 95%CI 0.88 to 6.76, p = 0.087), both compared with patients with no glucocorticoids. The rest of the model included the same variables than the previous analysis (Table 5). In the subcohort with low SpO2/FiO2, week-2-MP decreased mortality compared with no glucocorticoids, HR 0.14, 95%CI 0.02 to 1.28, p = 0.082. The remaining sub-groups showed non significant differences.

In the model with the combined death or intubation outcome, 2-week-MP were protective (HR 0.28, 95%CI 0.07 to 1.12, p = 0.072), whilst non-pulse glucocorticoids increased the risk

**Table 2. Predictors of death: Final models.**

| Variable | HR (95%CI) | p |
|---|---|---|
| WHOLE COHORT (n = 242) | | |
| Week-2-MP | 0.35 (0.11–1.06) | 0.064 |
| Non-pulse glucocorticoids | 3.01 (1.28–7.14) | 0.012 |
| Hypertension | 2.89 (0.91–9.17) | 0.072 |
| SpO2/FiO2 | 0.94 (0.91–0.98) | 0.001 |
| CURB65 | | |
| Low risk | Reference | |
| Intermediate risk | 1.64 (0.55–4.88) | 0.371 |
| High risk | 7.72 (2.56–23.26) | <0.001 |
| PATIENTS WITH SpO2/FiO2 ≤353 (n = 122) | | |
| Week-2-MP | 0.31 (0.08–1.12) | 0.073 |
| Non-pulse glucocorticoids | 2.98 (1.09–8.17) | 0.034 |
| Hypertension | 3.14 (0.84–11.75) | 0.089 |
| SpO2/FiO2 | 0.92 (0.87-0-97) | 0.002 |
| CURB65 | | |
| Low risk | Reference | |
| Intermediate risk | 1.48 (0.41–5.34) | 0.546 |
| High risk | 10.29 (2.72–38.94) | 0.001 |

Week-2-MP: methyl-prednisolone pulses in week 2. HR: hazard ratio; CI: confidence interval.

*HR for SpO2/FiO2 is change in hazard for each increase of 10 units in its value.

(HR 3.83, 95%CI 1.52 to 9.68, p = 0.004) vs. patients on no glucocorticoids. The results were similar in the subcohort with a low SpO2/FiO2: week-2-MP vs. no glucocorticoids, HR 0.20, 95%CI 0.04 to 1.00, p = 0.050; non-pulse glucocorticoids vs. no glucocorticoids, HR 3.06, 95% CI 1.06 to 8.85, p = 0.039 (Table 6).

None of the analysis in the subcohort with high SpO2/FiO2 at week 2 showed significant differences between the four therapeutic groups.

The results of the four-level glucocorticoid Cox analysis with 2-week-MP as the main comparator are presented in Table 7. Week-2-MP was superior to all the other therapies in order to prevent both main outcomes. The effects were more marked among patients with low SaO2/FiO2.

## Discussion

This study supports the utility of glucocorticoids to improve the outcome of patients with COVID-19 pneumonia. Glucocorticoid use, however, should not be indiscriminate, but rather restricted to patients with laboratory evidence of inflammation and progressing respiratory compromise, and best used as short-course pulse therapy (125–250 mg/d of methyl-prednisolone during 3 days) administered during the second week after the onset of symptoms, where the hyperinflammatory reaction takes usually place.

By the time the outbreak reached our region in late February 2020, the use of glucocorticoids was discouraged by the WHO [3]. This recommendation was based on a review of the clinical evidence of steroid therapy in other similar viral diseases, such as SARS-Cov-1, Middle East respiratory syndrome (MERS), influenza and respiratory syncytial virus, and on the idea that glucocorticoids could actually delay viral clearance [10]. However, an observational study of 201 patients with COVID-19 pneumonia admitted to Wuhan Jinyintan Hospital published

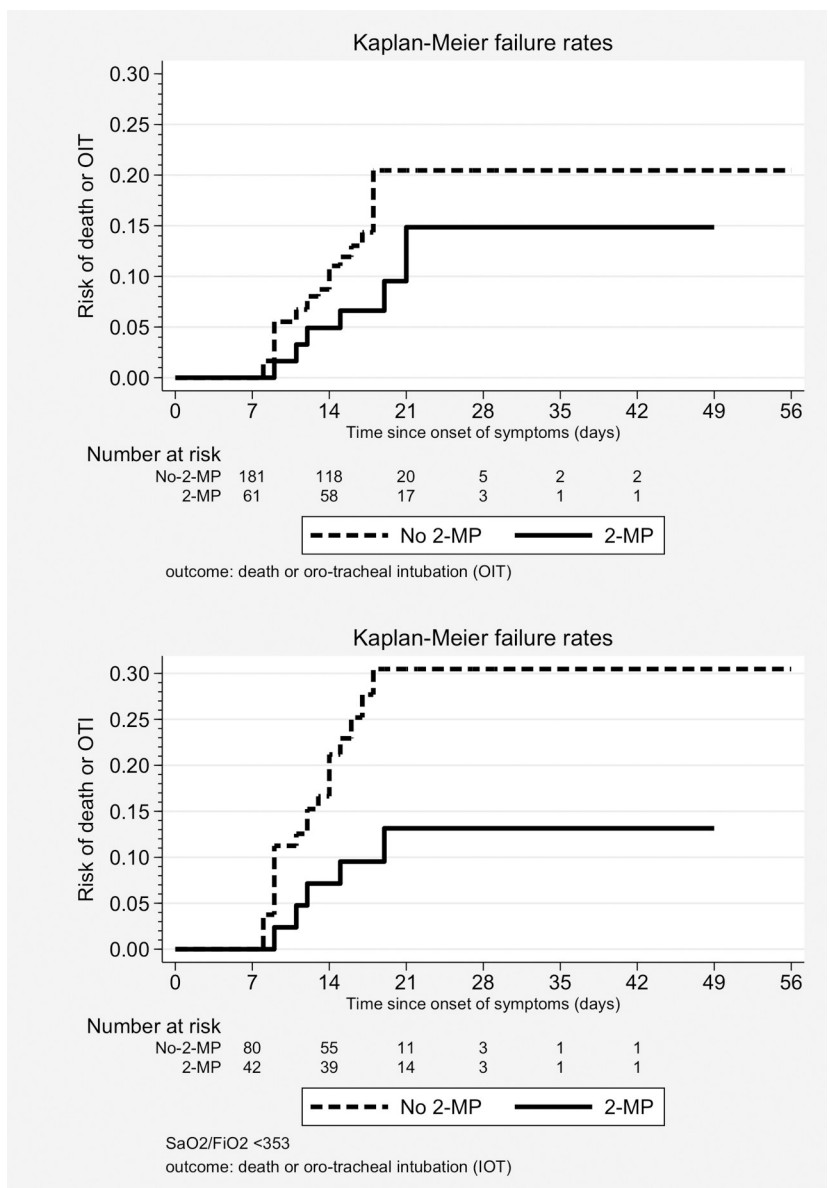

**Fig 2. Kaplan-Meier failure curves, second week methyl-prednisolone pulses (2-MP) vs. no 2-MP.** Outcome: death or intubation. (a) Whole cohort (n = 242). Log-rank test, p = 0.125. (b) Patients with low SpO2/FiO2 (n = 122). Log-rank test, p = 0.032.

in early March showed a reduced mortality among patients receiving methyl-prednisolone, although the dose, the duration and the time of administration were not specified [11].

Indeed, new insights into the pathogenesis of the disease seem to support immunoregulation of some kind in patients affected by SARS-Cov-2, with the recognition of an inflammatory phase of the disease that can lead to extensive lung and multisystemic injury [4]. This second phase usually takes place during the second week after the onset of symptoms.

Recent data also support the utility of glucocorticoids in COVID-19 disease [12–14]. In an observational study of 213 patients from Detroit [12], investigators compared patients of two groups, the standard of care (SOC) and early corticosteroid protocol groups (CP). Patients

**Table 3. Predictors of death or intubation: Final models.**

| Variable | HR (95%CI) | p |
|---|---|---|
| WHOLE COHORT (n = 242) | | |
| Week-2-MP | 0.33 (0.13–0.84) | 0.020 |
| Non-pulse glucocorticoids | 3.87 (1.87–8.02) | <0.001 |
| SpO2/FiO2 | 0.92 (0.89–0.94) | <0.001 |
| CURB65 | | |
| Low risk | Reference | |
| Intermediate risk | 1.93 (0.83–4.45) | 0.125 |
| High risk | 3.65 (1.48–9.02) | 0.005 |
| PATIENTS WITH SatO2/FiO2 ≤353 (n = 122) | | |
| Week-2-MP | 0.34 (0.12–0.94) | 0.038 |
| Non-pulse glucocorticoids | 3.16 (1.36–7.37) | 0.008 |
| SpO2/FiO2* | 0.88 (0.83-0-93) | <0.001 |
| CURB65 | | |
| Low risk | Reference | |
| Intermediate risk | 1.78 (0.69–4.57) | 0.234 |
| High risk | 3.94 (1.44–10.78) | 0.008 |

Week-2-MP: methyl-prednisolone pulses in week 2. HR: hazard ratio; CI: confidence interval.

*HR for SpO2/FiO2 is change in hazard for each increase of 10 units in its value.

were given a similar median of 40 mg/d of methyl-prednisolone for 3 days, however, the proportion of treated patients (SOC 56.5% vs. CP 68.2%) and the time from symptoms to methyl-prednisolone administration (SOC 10 days vs. CP 8 days) were different. The combined composite end point (admission to intensive care unit/mechanical ventilation/death) occurred in 54.3% vs. 34.9% (OR 0.45, 95%CI 0.26–0.79) of patients in the SOC and CP, respectively.

A second observational study from Madrid compared 396 patients admitted for COVID-19 treated with glucocorticoids with 67 not treated [13]. In the glucocorticoid-treated group, methyl-prednisolone was given either at 1 mg/kg/d (78.3%) or as 2 to 4-day pulses, up to 500 mg/d (21.7%) with subsequent tapering in 25% patients. The duration of therapy with 1 mg/kg/d was not specified. The median time from disease onset to initiation of glucocorticoid was 10 days. This study showed a significant reduction in mortality among glucocorticoid users (13.9% vs. 23.9%, HR 0.51, 95%CI0.27–0.96), with differences being significant only in the subgroup classified as moderate-severe disease. There was no difference in mortality between patients receiving 1 mg/Kg/d or pulses, however, 70 patients received MP a mean of 5 days after the failure of the 1 mg/kg/d scheme.

The preliminary results from the RECOVERY trial have just been released [15]. Patients with suspected or confirmed COVID-19 infection were randomised to receive dexamethasone (n = 2104) or SOC (n = 4321). The dexamethasone protocol consisted of 6 mg/d for up to 10 days, although the actual median number of days was 6, and 7% of SOC patients also received dexamethasone. The primary outcome, all-cause mortality within 28 days of randomization, was met by 21.6% patients allocated to dexamethasone vs. 24.6% allocated to usual care (RR 0.83; 95% CI 0.74 to 0.92). The reduction of mortality was only significant in patients receiving respiratory support. Following the press release of these results, the WHO has already demanded an increase in dexamethasone production [15].

In contrast, a more recent Brazilian double-blind, randomized, placebo-controlled trial in patients with clinical or radiological suspicion of COVID-19 pneumonia, with SpO2 ≤ 94% at

**Table 4. Clinical characteristics of the cohort according to type of glucocorticoid therapy.**

| VARIABLE | No glucocorticoids (n = 122) | Non-pulse-glucocorticoids (n = 36) | Out-of-week-2-MP (n = 30) | Week-2-MP (n = 54) | p-value |
|---|---|---|---|---|---|
| Age (years), mean (sd) | 62.9 (14.7) | 69.0 (13.1) | 63.9 (16.4) | 64.9 (12.5) | 0.159 |
| Male n (%) | 63 (51.6) | 29 (80.6) | 23 (76.7) | 35 (64.8) | **0.003** |
| Follow-up (days)*, mean (sd) | 15.9 (5.0) | 19.5 (10.7) | 20.2 (10.7) | 20.7 (5.9) | **<0.001** |
| Diabetes mellitus (n %) | 25 (20.9) | 10 (27.8) | 9 (30.0) | 7 (13.0) | 0.206 |
| Overweight n (%) | 29 (23.8) | 5 (13.9) | 5 (16.7) | 10 (18.5) | 0.536 |
| Hypertension n (%) | 51 (41.8) | 22 (61.1) | 15 (50.0) | 29 (53.7) | 0.165 |
| Chronic bronchopathy n (%) | 29 (23.8) | 15 (41.7) | 6 (20.0) | 12 (22.2) | 0.116 |
| Active cancer n (%) | 15 (12.3) | 5 (13.9) | 0 (0.0) | 6 (11.1) | 0.231 |
| Neurodegenerative disease n (%) | 3 (2.5) | 1 (2.8) | 2 (6.7) | 3 (5.6) | 0.601 |
| Autoimmune disorder n (%) | 4 (3.3) | 3 (8.3) | 0 (0.0) | 2 (3.7) | 0.339 |
| Immunosuppressive therapy n (%) | 10 (8.2) | 7 (19.4) | 0 (0.0) | 4 (7.4) | **0.041** |
| Curb65: high risk n (%) | 5 (4.1) | 6 (16.7) | 5 (16.7) | 3 (5.6) | **0.040** |
| Time of symptoms to admission (days) mean (sd) | 6.8 (3.1) | 5.4 (3.1) | 5.4 (3.6) | 7.6 (2.7) | **0.001** |
| Lymphocytes (count/mm$^3$) median (iqr) | 950 (540) | 670 (460) | 625 (500) | 690 (460) | **<0.001** |
| Platelets (count/mm$^3$) median (iqr) | 205,000 (116,000) | 212,000 (154,500) | 228,500 (135,000) | 214,500 (165,000) | 0.900 |
| Ferritin (mg/dl) median (iqr) | 402 (561) | 444 (920) | 1231 (1327) | 818 (919) | **<0.001** |
| D-dimers ng/ml median (iqr) | 410 (991) | 565 (1,115) | 750 (1670) | 525 (890) | 0.121 |
| C-reactive protein (mg/dl) median (iqr) | 60.1 (109.2) | 78.2 (97.5) | 110.8 (97.5) | 120.6 (94.7) | **0.001** |
| SaO2/FiO2 median (iqr) | 450.0 (118.1) | 445.5 (214.0) | 299.5 (344.2) | 330.5 (206.3) | **<0.001** |
| SaO2/FiO2 <353 n (%) | 43 (35.2) | 19 (52.8) | 21 (70.0) | 35 (64.8) | **<0.001** |
| Hydroxychloroquine n (%) | 109 (89.3) | 32 (88.9) | 29 (96.7) | 54 (100) | 0.227 |
| Days on hidroxicloroquina mean (sd) | 6.0 (2.9) | 6.8 (4.0) | 6.2 (2.7) | 6.9 (2.7) | 0.290 |
| Lopinavir-Ritonavir n (%) | 109 (89.3) | 31 (83.1) | 27 (90.0) | 51 (94.4) | 0.602 |
| Betaferon n (%) | 8 (6.6) | 3 (8.3) | 1 (3.3) | 1 (1.9) | 0.472 |
| LMWH n (%) | 110 (90.2) | 32 (88.9) | 30 (100) | 53 (98.2) | 0.076 |

Week-2-MP: methyl-prednisolone pulses in week 2; LMWH: low molecular weight heparin; sd: standard deviation; iqr: interquartile range.

* From disease onset to death, discharge or end of the study period.

**Table 5. Predictors of death: final models using the type of glucocorticoid therapy.**

| Variable | HR (95%CI) | p |
|---|---|---|
| WHOLE COHORT (n = 242) | | |
| Glucocorticoid therapy | | |
| No glucocorticoids | reference | |
| Non-pulse glucocorticoids | 2.44 (0.88–6.76) | 0.087 |
| Out-of-week-2-MP | 1.22 (0.33–4.42) | 0.767 |
| Week-2-MP | 0.24 (0.05–1.24) | 0.088 |
| Hypertension | 2.96 (0.92–9.56) | 0.069 |
| SpO2/FiO2* | 0.94 (0.91–0.97) | 0.001 |
| CURB65 | | |
| Low risk | reference | |
| Intermediate risk | 1.75 (0.58–5.21) | 0.318 |
| High risk | 7.66 (2.46–28.85) | <0.001 |
| PATIENTS WITH SpO2/FiO2 ≤353 (n = 122) | | |
| Glucocorticoid therapy | | |
| No glucocorticoids | reference | |
| Non-pulse glucocorticoids | 2.30 (0.71–7.47) | 0.169 |
| Out-of-week-2-MP | 1.40 (0.35–5.54) | 0.636 |
| Week-2-MP | 0.14 (0.02–1.28) | 0.082 |
| Hypertension | 3.11 (0.78–12.41) | 0.108 |
| SaO2/FiO2* | 0.91 (0.86–0.96) | 0.001 |
| CURB65 | | |
| Low risk | reference | |
| Intermediate risk | 1.67 (0.45–6.18) | 0.441 |
| High risk | 10.85 (2.58–45.56) | 0.001 |

MP: methyl-prednisolone pulses; Week-2-MP: methyl-prednisolone pulses in week 2. HR: hazard ratio; CI: confidence interval.

*HR here estimates change in hazard by 10 units increase in SpO2/FiO2.

room air or need for non-invasive or invasive ventilation reported no benefit from the use of methyl-prednisolone [16]. They were treated with methyl-prednisolone, 0.5 mg/kg/12h, or placebo for five days. The 28-day mortality rates in the 393 patients who completed the follow-up was 37.1% vs. 38.2% in the intervention and placebo groups, respectively. Of note, the median number of days between disease onset and randomization was 13 days in both groups. Authors reported, in the subgroup of patients older than 60 years who also had higher levels of C-reactive protein, a lower mortality among patients treated with methyl-prednisolone.

Thus, a beneficial role of glucocorticoids in patients with severe COVID-19 disease has been generally found [12–14, 16]. Whether they should be used in short-term pulses (i.e. doses of methylprednisolone between 100–500 mg/d for 3–4 days) or at lower doses for more prolonged periods of time is still an open question. The advantages of dexamethasone over methyl-prednisolone are not well established; besides the large sample size and the randomised controlled trial design of the RECOVERY study [14], the difference between the treated and untreated groups was quantitatively smaller than in the studies using methyl-prednisolone [12, 13]. The time of administration of glucocorticoids has been variable, although in the Brazilian study, which concluded with mostly negative results, they were administered within the third week of disease in 50% of patients [16]. Thus, specific recommendations about which corticosteroid should be administered, at what dose, for how long and when in the clinical course, could not be yet made.

**Table 6. Predictors of death or intubation.** Final models using the type of glucocorticoid therapy.

| Variable | HR (95%CI) | p |
|---|---|---|
| WHOLE COHORT (n = 242) | | |
| Glucocorticoid therapy | | |
| No glucocorticoids | reference | |
| Non-pulse glucocorticoids | 3.83 (1.51–9.68) | 0.004 |
| Out-of-week-2-MP | 2.06 (0.71–6.00) | 0.183 |
| Week-2-MP | 0.28 (0.07–1.12) | 0.072 |
| SpO2/FiO2* | 0.92 (0.89–0.95) | <0.001 |
| CURB65 | | |
| Low risk | reference | |
| Intermediate risk | 2.38 (1.00–5.65) | 0.049 |
| High risk | 3.98 (1.58–9.89) | 0.003 |
| PATIENTS WITH SpO2/FiO2 ≤353 (n = 122) | | |
| Glucocorticoid therapy | | |
| No glucocorticoids | reference | |
| Non-pulse glucocorticoids | 3.06 (1.06–8.85) | 0.039 |
| Out-of-week-2-MP | 1.92 (0.61–6.00) | 0.263 |
| Week-2-MP | 0.20 (0.04–1.00) | 0.050 |
| SpO2/FiO2* | 0.88 (0.83–0.93) | <0.001 |
| CURB65 | | |
| Low risk | reference | |
| Intermediate risk | 2.49 (0.91–6.80) | 0.075 |
| High risk | 5.00 (1.73–14.47) | 0.003 |

MP: methyl-prednisolone pulses; Week-2-MP: methyl-prednisolone pulses in week 2. HR: hazard ratio; CI: confidence interval.

*HR here estimates change in hazard by 10 units increase in SaO2/FiO2.

Our study confirms that patients with COVID-19 disease with a high inflammatory profile and respiratory compromise are most likely to benefit from glucocorticoid therapy. In addition, our study found that only in patients treated in the second week after the onset of symptoms seemed MP to be effective. We also observed a clear difference between the use of a short course of MP and more prolonged reduced doses of glucocorticoids, the former preventing and the second being associated to an increased risk of both unfavourable outcomes in all developed models. The presence of unidentified sources of bias cannot be excluded even after

**Table 7. Cox models for death and death or intubation by type of glucocorticoid therapy.**

| | Death | | | | Death or intubation | | | |
|---|---|---|---|---|---|---|---|---|
| | Whole cohort | | SpO2/FiO2 ≤353 | | Whole cohort | | SpO2/FiO2 ≤353 | |
| | HR (95%CI)* | p | HR (95%CI)* | p | HR (95%CI)** | p | HR (95%CI)** | p |
| Week-2-MP | reference | | reference | | reference | | reference | |
| Non-pulse glucocorticoids | 10.09 (2.14–47.50) | 0.003 | 16.20 (1.86–141.05) | 0.012 | 13.71 (3.76–50.07) | <0.001 | 15.50 (3.14–76.38) | 0.001 |
| Out-of-week-2-MP | 5.04 (0.91–27.86) | 0.064 | 9.83 (0.78–12.41) | 0.044 | 7.38 (1.86–29.29) | 0.004 | 9.72 (1.91–49.31) | 0.006 |
| No glucocorticoids | 4.14 (0.81–21.23) | 0.088 | 7.04 (0.78–63.71) | 0.082 | 3.58 (0.89–14.32) | 0.072 | 5.07 (1.00–25.67) | 0.050 |

MP: methyl-prednisolone pulses; Week-2-MP: methyl-prednisolone pulses in week 2. HR: hazard ratio; CI: confidence interval.

*Model adjusted by hypertension, SaO2FiO2 and Curb65 score.

**Model adjusted by SaO2FiO2 and Curb65 score.

statistical adjustment for other prognostic factors and thus we cannot assure this deleterious effect on non-pulse methyl-prednisolone, moreover in the light of the results of previous studies. However, the superiority of week-2-MP over all the remaining options was clearly identified in our cohort. Our study does not support the use of non-pulse glucocorticoids in the setting of severe COVID-19 pneumonia, either alone or combined with MP.

These findings are actually in agreement with the results seen in patients with systemic lupus erythematosus and other autoimmune diseases. The activation of the non-genomic way by MP, given during a short period of time, maximises the anti-inflammatory activity of glucocorticoids without relevant side effects [17–19]. On the other hand, doses in the range of 40–90 mg/d during a period of one week or more are less effective and can increase the risk of infections. The reason for this is the full activation of the genomic way, which is responsible for most of the serious secondary effects of glucocorticoids [17–19].

## Limitations

The most important limitation of this study is the observational design with a high variability in the therapeutic schemes, a consequence of the rapidly changing situation during this highly stressing pandemic. Indeed, the actual number of patients treated with MP was lower than expected after the approval of the revised protocol. Patients receiving MP also had a more severe disease. Thus, although the analysis included a high number of potentially predictive variables in the initial multivariate Cox models, it is possible that some unidentified confounder may have influenced our results. We also had a low number of patients meeting the final endpoints compared with other studies [12–14, 16]. A possible explanation for this is that, despite the high pressure attained, our hospital did not collapse as many others in Spain. Despite the consequent limited statistical power, the risk reduction among patients receiving week-2-MP was consistently between 60–70% for both endpoints in all the models, which supports a clinically relevant effect.

A critical point in the clinical application of our results is the correct identification of the point of disease onset, in order to make the correct evaluation and starting MP, if needed, early in the second week. This is not always easy, since non-specific symptoms may precede the fully symptomatic phase of the disease by several days and because some patients even with radiological features of severe pneumonia can be oligosymptomatic [20]. Thus, a thorough anamnesis must be made at admission with specific questions regarding the starting date of fever, persistent cough and/or dyspnea.

## Conclusion

This study confirms that MP, 125–250 mg/d for 3 consecutive days given during the second week of disease without subsequent tapering, improve the prognosis of patients with COVID-19 pneumonia, features of inflammatory activity and respiratory deterioration. Our results open the door to a more rational and planned management of patients with COVID-19. Those with negative inflammatory markers and normal SpO2 seem to have a good prognosis, thus clinical observation monitoring oxygen saturation (even at home), could be appropriate. On the other hand, those approaching the second week of disease with worsening inflammatory markers and respiratory failure would greatly benefit from a short course of MP, which could not only be life-saving, but also help avoid the overload of critical care units.

## Supporting information

**S1 Dataset. COVID pulses.**
(XLS)

## Acknowledgments

We thank all the health professionals from Hospital Universitario Cruces involved in the care of patients with COVID-19.

The Cruces COVID Study Group is formed by:

Service of Internal Medicine. Hospital Universitario Cruces:

Jokin Dominguez

Luis Dueña

Susanna Dunder

Gorka de Frutos

Ignacio Fernandez-Huerta

Jose-Gabriel Erdozain

Mikel Escalante

Juan-Manuel Goiri

Carlota Gomez-Carballo

Nuria López-Osle

Agustin Martinez-Berriotxoa

Juan Monte

Jose Rodriguez-Chinesta

Alejandro Rodrigo

Guillermo Ruiz-Irastorza

Angel Sebastian

Adriana Soto

Amaia Ugarte

Jimena Varona

Laura Velasco

Irama Villar

Infectious Diseases Unit. Hospital Universitario Cruces:

Elena Bereciartua

Maria-Jose Blanco

Mikel del Alamo

Gorane Euba

Josune Goikoetxea

Laura Guio

Marta Ibarrola

Javier Nieto

Regino Rodriguez-Alvarez

## Author Contributions

**Conceptualization:** Guillermo Ruiz-Irastorza, Jose-Ignacio Pijoan, Elena Bereciartua, Amaia Ugarte, Agustin Martinez-Berriotxoa.

**Data curation:** Elena Bereciartua, Susanna Dunder, Jokin Dominguez, Paula Garcia-Escudero, Alejandro Rodrigo, Carlota Gomez-Carballo, Jimena Varona, Laura Guio, Marta Ibarrola.

**Formal analysis:** Guillermo Ruiz-Irastorza, Jose-Ignacio Pijoan.

**Methodology:** Guillermo Ruiz-Irastorza.

**Supervision:** Guillermo Ruiz-Irastorza.

**Writing – original draft:** Guillermo Ruiz-Irastorza.

**Writing – review & editing:** Guillermo Ruiz-Irastorza, Jose-Ignacio Pijoan, Elena Bereciartua, Susanna Dunder, Jokin Dominguez, Paula Garcia-Escudero, Alejandro Rodrigo, Carlota Gomez-Carballo, Jimena Varona, Laura Guio, Marta Ibarrola, Amaia Ugarte, Agustin Martinez-Berriotxoa.

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
