## [Decision Letter · Decision Letter 0]

20 Aug 2020

PONE-D-20-23728

SECOND WEEK METHYL-PREDNISOLONE PULSES IMPROVE PROGNOSIS IN PATIENTS WITH SEVERE CORONAVIRUS DISEASE 2019 PNEUMONIA: AN OBSERVATIONAL COMPARATIVE STUDY USING ROUTINE CARE DATA.

PLOS ONE

Dear Dr. Ruiz-Irastorza,

Thank you for submitting your manuscript to PLOS ONE. After careful consideration, we feel that it has merit but does not fully meet PLOS ONE’s publication criteria as it currently stands. Therefore, we invite you to submit a revised version of the manuscript that addresses the points raised during the review process.

We look forward to receiving your revised manuscript.

Kind regards,

Aleksandar R. Zivkovic

Academic Editor

PLOS ONE

Journal Requirements:

2. Please include the date(s) on which you accessed the databases or records to obtain the data used in your study.

3. Thank you for stating in the text of your manuscript "The protocol was approved by the Basque Country Research Ethics Committee (code EPA2020032) in accordance with the Declaration of Helsinki’s guidelines for research in humans. This study has been registered in the EU PAS Register with the number EUPAS36287.". Please also add this information to your ethics statement in the online submission form.

4. In your ethics statement in the Methods section and in the online submission form, please provide additional information about the data used in your retrospective study. Specifically, please ensure that you have discussed whether all data were fully anonymized before you accessed them and/or whether the IRB or ethics committee waived the requirement for informed consent. If patients or next of kin provided informed written consent to have data from their medical records used in research, please include this information.

5. One of the noted authors is a group or consortium [Cruces COVID Study Group]. In addition to naming the author group, please list the individual authors and affiliations within this group in the acknowledgments section of your manuscript. Please also indicate clearly a lead author for this group along with a contact email address.

Reviewers' comments:

Reviewer's Responses to Questions

**Comments to the Author**

1. Is the manuscript technically sound, and do the data support the conclusions?

Reviewer #1: Partly

Reviewer #2: Partly

2. Has the statistical analysis been performed appropriately and rigorously? 

Reviewer #1: I Don't Know

Reviewer #2: I Don't Know

3. Have the authors made all data underlying the findings in their manuscript fully available?

Reviewer #1: No

Reviewer #2: No

4. Is the manuscript presented in an intelligible fashion and written in standard English?

Reviewer #1: Yes

Reviewer #2: Yes

5. Review Comments to the Author

Reviewer #1: This study further supports the findings of recently published work evaluating the impact of steroids in patients hospitalised with COVID19 pneumonia. The concordance of the results presented here reinforce the evidence base, which although reassuring, does not really add anything to what has already been established in the RECOVERY trial. Indeed, the additional benefit of a small retrospective observational study compared to a large, prospective clinical trial that has clearly established the efficacy of steroids in patients with severe disease is questionable.

Issues:

Observational, retrospective nature of study and relatively small numbers (242 vs 6425 in RECOVERY). MP was given at the discretion of the attending physician and this is likely to have introduced bias. The characteristics of patients who received MP compared to those who did not differed making it hard to draw conclusions from the outcomes – for example, only 14.8% who had MP had diabetes compared to 23.2% of those who did not. Patients receiving MP had higher inflammatory markers (ferritin, CRP), and more were on LMWH, which may be of significance given the high incidence of pulmonary emboli in patients with acute infection. Table 2 compares the demographics of patients who received week 2 MP with those who did not but the latter group also includes those who had MP at some other point (33 patients). It would be useful to separate out these groups and to provide further information on the 14% of patients who had MP outside of week 2.

The secondary outcome was nosocomial infections but it is unclear how these were defined (and may have be difficult to differentiate from worsening primary disease). Details of the steroid regime and characteristics of patients receiving non-pulsed GC were also not clearly stated and there appears to be a wide dose range (could some patients be on pre-existing steroids?). The authors mention that pulsed MP was not associated with increased infection rate whereas non-pulsed GCs were. The numbers of patients in the GC subgroup was small and so I do not think conclusions can be drawn from the results. Furthermore, given that the duration of steroids was relatively short in both groups (only a mean of 6.5 days in the non-pulsed GC group), it would seem unlikely that infection risk would be significantly higher in the group receiving lower dose for marginally longer. I suspect that confounding factors are responsible but no information has been given on the non-pulsed GC subgroup. Whilst the authors confidently report that non-pulsed GC is associated with higher infection risk, they do not provide a possible explanation for their observation and place too much weight on this finding.

A lot of emphasis was put on timing of MP and the importance of short duration but little information was given regarding when patients had it outside this window i.e. how many had it at week 1 and how many at week 3. As the majority of patients who present to hospital have had symptoms for a number of days (often a week or so), then most would not have received it within the first week. Patients receiving MP in the third week, at the judgement of the attending physician, are likely to have a severe protracted form of the disease and therefore difficult to determine whether the steroids really were harmful (again, a separate demographic table looking at this group of patients would be helpful). The majority of patients in the RECOVERY trial received dexamethasone in the second week so this is not a new finding.

The first paragraph of the introduction states that no drug has been found to improve the outcome of severe COVID-19 infection. Clearly, this is no longer true with the publication of the RECOVERY trial (and other smaller studies), which are then referred to in the discussion.

The rationale for high-dose steroids for a short period of time is logical in the context of severe COVID19 infection where an exuberant immune response is thought to drive the later phase pathophysiology, although the authors mention that this is the case in many autoimmune diseases. This is not true however, and in many diseases (i.e. inflammatory interstitial lung disease (including organising pneumonia), vasculitis, rheumatoid arthritis etc) require long term immunosuppression which often takes the form of a high dose steroid regime that is weaned over time.

In summary, the data presented in this study is not new and the findings have recently published elsewhere. The secondary conclusions regarding infection risk and timing of MP are not adequately supported by the data presented.

Reviewer #2: PONE-D-20-23728

The authors conducted an observational study in which patients with COVID-19 pneumonia who received methyl-prednisolone pulse therapy during the second week of disease (week-2-MP) are compared with those who did not receive, in terms of time to death, and time to death or endotracheal intubation. They concluded that week-2-MP are effective in improving the prognosis of patients with severe COVID-19 pneumonia.

The authors do not set a significance level based on the p-value of the statistical analysis. Results are arbitrarily described to support the author's hypothesis. This study suggests that methyl-prednisolone pulse therapy may improve the prognosis of CODIV-19 patients depending on the timing of administration and the patient's condition. On the other hand, this study shows that non-pulse glucocorticoid treatment, and methyl-prednisolone pulse therapy other than the second week were harmful. In both respects, this study may be helpful for broad readers to treat COVID-19 patients, while several undescribed matters of the study should be clarified.

# The reviewer is not provided with supplemental table 1, 2 and 3, if any.

Questions;

1. The definition of pneumonia in this study is not described. How did the authors diagnose pneumonia, by chest CT scan, chest radiograph, or clinical impression?

2. The definition of onset of symptoms is not described. What are the symptoms that represent the onset of COVID-19 in this study?

3. Does “SaO2” in this manuscript represent arterial oxygen saturation?

Is this SpO2 (percutaneous arterial oxygen saturation), isn’t it?

4. How was the FiO2 for each patient who did not receive endotracheal intubation determined?

Was it really measured, or was it only speculated?

5. “inflammatory state at admission was defined as the presence of any of the following: lymphocyte count <800/mm3, platelet count <150,000/mm3, ferritin >1000 mg/dl, C-reactive protein >100 mg/dl, D-dimers >1000 ng/ml. “

The values of ferritin and C-reactive protein in this definition seem to be very high. What are the normal ranges of these five measurements at the hospital where this study was done?

6. page 9; “The Kaplan-Meier failure curves (figure 1a) showed a decreased risk of death of patients in the week-2-MP group with a trend for significance (log-rank test, p=0.102).”

How do the authors select significant results from this study based on the statistical p-value?

What does “a trend for significance” mean?

7. Figure 1b has no title, like "2b: Patients with low SaO2/FiO2 (n= 122). Log-rank test, p=0.032.”.

8. page 10 “In the model with mortality as outcome, week-2-MP patients showed a trend for lower mortality (HR 0.48, 95%CI 0.14 to 1.57, p=0.225), whilst out-of-week-2- MP patients had an increased risk of death (HR 2.49, 95%CI 0.87 to 7.11, p= 0.088), both compared with no MP patients.“

Comparison among 2-week-MP, out-of-week-2-MP and no MP patients is critical for this study. These data (probably in supplemental table 1, 2 and 3) should be presented as main tables.

9. Whole cohort (n=242) may contain the following four patient groups.

1, Week-2-MP patients (n=61).

2, Out-of-week-2-MP patients (week 1 or 3, n=33?).

3, Patients who did not receive any MP, but received any glucocorticoid (n=35?).

4, Patients who did not receive any glucocorticoid (n=113?).

The authors do not present the outcomes of these four patient categories clearly. Tables 2 and 3 should include HRs for these four patient groups. What is the outcome of patients who did not receive any glucocorticoid in Table 2 and 3 analysis?

10. Since out-of-week-2-MP patients had an increased risk of death, awareness of the onset of disease is critical. Actually, most people with SARS-CoV-2 infection have no recognized symptoms. Nevertheless, some of those people have asymptomatic pneumonia diagnosed by lung CT scan (CT features of SARS-CoV-2 pneumonia according to clinical presentation: a retrospective analysis of 120 consecutive patients from Wuhan city. Eur. Radiol. 30 (8), 4417–4426), and have even asymptomatic hypoxemia. It may be difficult to determine the actual onset of COVID-19 pneumonia. Therefore, clinically, it may be difficult to decide when the second week of disease begins and when week-2-MP therapy should be given. How do the authors think about it?

6. PLOS authors have the option to publish the peer review history of their article (what does this mean?). If published, this will include your full peer review and any attached files.

Reviewer #1: No

Reviewer #2: **Yes: **Tomohiko Aoe

---

## [Author Response · Author response to Decision Letter 0]

3 Sep 2020

RESPONSE TO REVIEWERS.

-We thank the reviewers for their comments. We feel that after the revision following their suggestions the study has improved in consistency and clarity of the results. 

Reviewer #1: This study further supports the findings of recently published work evaluating the impact of steroids in patients hospitalised with COVID19 pneumonia. The concordance of the results presented here reinforce the evidence base, which although reassuring, does not really add anything to what has already been established in the RECOVERY trial. Indeed, the additional benefit of a small retrospective observational study compared to a large, prospective clinical trial that has clearly established the efficacy of steroids in patients with severe disease is questionable.

-We agree that the design and the size of our study are inferior to RECOVERY. However, we honestly feel that our work adds important complementary information regarding the use of corticosteroids in these patients: it supports the effective use of methyl-prednisolone, as an alternative to dexamethasone (opening the range of effective medications can be crucial in case of shortage of widely used drugs, as could be the future situation), of pulses instead of mg/kg/d doses, taking advantage of the non-genomic way activation and of short-term courses, potentially decreasing toxicity. This manuscript also contributes to answer two key related questions: when and to which patients should this therapy be given. 

Issues:

Observational, retrospective nature of study and relatively small numbers (242 vs. 6425 in RECOVERY). MP was given at the discretion of the attending physician and this is likely to have introduced bias. The characteristics of patients who received MP compared to those who did not differed making it hard to draw conclusions from the outcomes – for example, only 14.8% who had MP had diabetes compared to 23.2% of those who did not. Patients receiving MP had higher inflammatory markers (ferritin, CRP), and more were on LMWH, which may be of significance given the high incidence of pulmonary emboli in patients with acute infection. Table 2 compares the demographics of patients who received week 2 MP with those who did not but the latter group also includes those who had MP at some other point (33 patients). It would be useful to separate out these groups and to provide further information on the 14% of patients who had MP outside of week 2.

-We deeply thank this comment. According to this remark and to reviewer 2 comments, we have added a more detailed analysis of patients within the four different therapeutic subgroups: no glucocorticoids; non-pulse glucocorticoids; week 2 MP and out of week 2 MP. We believe that with these results the final message of the study is reinforced.

The secondary outcome was nosocomial infections but it is unclear how these were defined (and may have be difficult to differentiate from worsening primary disease). Details of the steroid regime and characteristics of patients receiving non-pulsed GC were also not clearly stated and there appears to be a wide dose range (could some patients be on pre-existing steroids?). The authors mention that pulsed MP was not associated with increased infection rate whereas non-pulsed GCs were. The numbers of patients in the GC subgroup was small and so I do not think conclusions can be drawn from the results. Furthermore, given that the duration of steroids was relatively short in both groups (only a mean of 6.5 days in the non-pulsed GC group), it would seem unlikely that infection risk would be significantly higher in the group receiving lower dose for marginally longer. I suspect that confounding factors are responsible but no information has been given on the non-pulsed GC subgroup. Whilst the authors confidently report that non-pulsed GC is associated with higher infection risk, they do not provide a possible explanation for their observation and place too much weight on this finding.

-We agree with the reviewer that that we have attributed too much weight to infections. The diagnosis of the clinician was assumed by the investigators and we do not have much data to refine this. We agree that probably a number of confounders are playing a relevant role. After revising the data and the analysis, we are convinced that this section adds more shadows than light to the study, and, not being an essential research question, we have decided to drop this section. We thank the reviewer for his comments. 

A lot of emphasis was put on timing of MP and the importance of short duration but little information was given regarding when patients had it outside this window i.e. how many had it at week 1 and how many at week 3. As the majority of patients who present to hospital have had symptoms for a number of days (often a week or so), then most would not have received it within the first week. Patients receiving MP in the third week, at the judgement of the attending physician, are likely to have a severe protracted form of the disease and therefore difficult to determine whether the steroids really were harmful (again, a separate demographic table looking at this group of patients would be helpful). The majority of patients in the RECOVERY trial received dexamethasone in the second week so this is not a new finding.

-We have added information regarding the out of week 2 MP group. The new by groups of glucocorticoid administration analysis shows that MP out of week two do not seem to be effective but do not confer a worse prognosis if not given with additional non-pulse glucocorticoids (see tables 5 and 6). However, we believe that the message that MP should be given during the second week in order to be beneficial is important, also suggested by RECOVERY and supported by current pathophysiological understanding. 

The first paragraph of the introduction states that no drug has been found to improve the outcome of severe COVID-19 infection. Clearly, this is no longer true with the publication of the RECOVERY trial (and other smaller studies), which are then referred to in the discussion.

-Thank you for the comment, this sentence has been deleted.

The rationale for high-dose steroids for a short period of time is logical in the context of severe COVID19 infection where an exuberant immune response is thought to drive the later phase pathophysiology, although the authors mention that this is the case in many autoimmune diseases. This is not true however, and in many diseases (i.e. inflammatory interstitial lung disease (including organising pneumonia), vasculitis, rheumatoid arthritis etc) require long term immunosuppression which often takes the form of a high dose steroid regime that is weaned over time.

-The practice in our Autoimmune Diseases Unit is avoiding high-dose glucocorticoids and limiting the time on prednisone >5 mg/d, precisely by using MP as one of the main resources. We have large published experience on that, mainly on lupus, showing the efficacy and lack of toxicity of this approach. This has been summarised in reference 17.

In summary, the data presented in this study is not new and the findings have recently published elsewhere. The secondary conclusions regarding infection risk and timing of MP are not adequately supported by the data presented.

-We hope that after the modifications the reviewer feels happier with our study and considers that it contains useful information for treatment of forthcoming patients with COVID-19 pneumonia.

Reviewer #2: PONE-D-20-23728

The authors conducted an observational study in which patients with COVID-19 pneumonia who received methyl-prednisolone pulse therapy during the second week of disease (week-2-MP) are compared with those who did not receive, in terms of time to death, and time to death or endotracheal intubation. They concluded that week-2-MP are effective in improving the prognosis of patients with severe COVID-19 pneumonia.

The authors do not set a significance level based on the p-value of the statistical analysis. Results are arbitrarily described to support the author's hypothesis. This study suggests that methyl-prednisolone pulse therapy may improve the prognosis of CODIV-19 patients depending on the timing of administration and the patient's condition. On the other hand, this study shows that non-pulse glucocorticoid treatment, and methyl-prednisolone pulse therapy other than the second week were harmful. In both respects, this study may be helpful for broad readers to treat COVID-19 patients, while several undescribed matters of the study should be clarified.

# The reviewer is not provided with supplemental table 1, 2 and 3, if any.

-Supplemental tables have been modified and included as table 4, 5 and 6.

Questions;

1. The definition of pneumonia in this study is not described. How did the authors diagnose pneumonia, by chest CT scan, chest radiograph, or clinical impression?

-Pneumonia was diagnosed by chest X-ray and, in some cases, by CT scan. All patients had a confirmation of the diagnosis by image informed by skilled radiologists. .

2. The definition of onset of symptoms is not described. What are the symptoms that represent the onset of COVID-19 in this study?

-The clinical protocol in our hospital included the time of onset of symptoms in the admission clinical record. We considered fever and/or persistent cough as the clinically meaningful presenting symptoms. 

3. Does “SaO2” in this manuscript represent arterial oxygen saturation?

Is this SpO2 (percutaneous arterial oxygen saturation), isn’t it?

-OK. Changed to SpO2. 

4. How was the FiO2 for each patient who did not receive endotracheal intubation determined?

Was it really measured, or was it only speculated?

-It was estimated according to the flow rates of the Venturi masks.

5. “inflammatory state at admission was defined as the presence of any of the following: lymphocyte count <800/mm3, platelet count <150,000/mm3, ferritin >1000 mg/dl, C-reactive protein >100 mg/dl, D-dimers >1000 ng/ml. “

The values of ferritin and C-reactive protein in this definition seem to be very high. What are the normal ranges of these five measurements at the hospital where this study was done?

-This definition of a high inflammatory state was used to select patients candidate to MP in the therapeutic protocol of our hospital. It was an empirical definition, in which the levels of ferritin, CRP and D-dimers well above the upper normal limit were set taking into account the marked elevation in such parameters seen in many patients with COVID-19 pneumonia. Since this definition was used to guide the clinical indication for MP, we decided to keep it unchanged for the study. The prognosis of admitted patients who had a negative inflammatory profile actually proved to be very good, with only two deaths in 91 patients and no intubations. Thus, we decided to keep this definition for the purposes of the study and to limit the analysis to those patients with an inflammatory profile, where a decision about how to deal with their impending clinical deterioration had to be made.

6. page 9; “The Kaplan-Meier failure curves (figure 1a) showed a decreased risk of death of patients in the week-2-MP group with a trend for significance (log-rank test, p=0.102).”

How do the authors select significant results from this study based on the statistical p-value?

What does “a trend for significance” mean?

-We have substituted the term “a trend for significance” by “non significant”, 

7. Figure 1b has no title, like "2b: Patients with low SaO2/FiO2 (n= 122). Log-rank test, p=0.032.”.

-Thank you, the title has been added.

8. page 10 “In the model with mortality as outcome, week-2-MP patients showed a trend for lower mortality (HR 0.48, 95%CI 0.14 to 1.57, p=0.225), whilst out-of-week-2- MP patients had an increased risk of death (HR 2.49, 95%CI 0.87 to 7.11, p= 0.088), both compared with no MP patients.“

Comparison among 2-week-MP, out-of-week-2-MP and no MP patients is critical for this study. These data (probably in supplemental table 1, 2 and 3) should be presented as main tables.

9. Whole cohort (n=242) may contain the following four patient groups.

1, Week-2-MP patients (n=61).

2, Out-of-week-2-MP patients (week 1 or 3, n=33?).

3, Patients who did not receive any MP, but received any glucocorticoid (n=35?).

4, Patients who did not receive any glucocorticoid (n=113?).

The authors do not present the outcomes of these four patient categories clearly. Tables 2 and 3 should include HRs for these four patient groups. What is the outcome of patients who did not receive any glucocorticoid in Table 2 and 3 analysis?

-We thank very much the reviewer for this useful comment. We have modified the analysis of the therapeutic subcohorts by dividing patients into 4 groups. As written in the text, the four groups were: “a) no-glucocorticoids, i.e. patients not receiving glucocorticoids in any form (n=122); b) non-pulse glucocorticoids, i.e. patients receiving glucocorticoids at doses lower than 100 mg/d for periods longer than 3 days (n=36, with 10 of them also receiving pulses); c) out-of-week-2-MP, i.e. MP at week 1 or 3, with no additional glucocorticoids at lower doses (n=30); d) week-2-MP, i.e. patients receiving MP during week 2, with no additional glucocorticoids at lower doses (n=54). As our proposed schedule consists of MP with no following tapering scheme, we decided to group all patients receiving non-pulse courses of glucocorticoids for longer than 3 days into the same category, keeping in the MP groups those patients receiving only pulses.”.

10. Since out-of-week-2-MP patients had an increased risk of death, awareness of the onset of disease is critical. Actually, most people with SARS-CoV-2 infection have no recognized symptoms. Nevertheless, some of those people have asymptomatic pneumonia diagnosed by lung CT scan (CT features of SARS-CoV-2 pneumonia according to clinical presentation: a retrospective analysis of 120 consecutive patients from Wuhan city. Eur. Radiol. 30 (8), 4417–4426), and have even asymptomatic hypoxemia. It may be difficult to determine the actual onset of COVID-19 pneumonia. Therefore, clinically, it may be difficult to decide when the second week of disease begins and when week-2-MP therapy should be given. How do the authors think about it?

-Once the effect of non-pulse glucocorticoids has been separated from MP, out of week 2 MP have no longer a significant detrimental effect, although the trend is towards a worse prognosis than no-glucocorticoids and week-2-MP. We agree with the reviewer that sometimes it is not easy to establish the clinical onset of the disease. However, as supported by these and other data and also by the physiopathology of the disease, every effort must be made to identify the starting date in order to assure that MP are not given too early (during the viral phase, in which it could reduce the clearance of the virus) or too late (when the inflammatory damage is already made). We have added a comment on this and the reference given by the reviewer. 

-In addition to the changes made in response to the reviewers’ comments, we have introduced some modifications in the discussion. The reference to the GLUCOCOVID study has been deleted, since the study is not yet published since the outprint was launched in June and because the clinical management and outcomes of patients were confusing. Instead, we have added a reference to a recently published Brazilian RCT (METCOVID, reference 16).

---

## [Editor Report · Decision Letter 1]

7 Sep 2020

SECOND WEEK METHYL-PREDNISOLONE PULSES IMPROVE PROGNOSIS IN PATIENTS WITH SEVERE CORONAVIRUS DISEASE 2019 PNEUMONIA: AN OBSERVATIONAL COMPARATIVE STUDY USING ROUTINE CARE DATA.

PONE-D-20-23728R1

Dear Dr. Ruiz-Irastorza,

We’re pleased to inform you that your manuscript has been judged scientifically suitable for publication and will be formally accepted for publication once it meets all outstanding technical requirements.

Kind regards,

Aleksandar R. Zivkovic

Academic Editor

PLOS ONE
---

## [Editor Report · Acceptance letter]

11 Sep 2020

PONE-D-20-23728R1 

Second week methyl-prednisolone pulses improve prognosis in patients with severe coronavirus disease 2019 pneumonia: an observational comparative study using routine care data. 

Dear Dr. Ruiz-Irastorza:

I'm pleased to inform you that your manuscript has been deemed suitable for publication in PLOS ONE. Congratulations! Your manuscript is now with our production department. 

Kind regards, 

on behalf of

Dr. Aleksandar R. Zivkovic 

Academic Editor

PLOS ONE